

# Application of bagging, boosting and stacking ensemble and EasyEnsemble methods to landslide susceptibility mapping in the Three Gorges Reservoir area of China

Xueling Wu [a, *], Junyang Wang [a]

[a] Institute of Geophysics and Geomatics, China University of Geosciences, Wuhan 430074, China

[*] Corresponding author.

*E-mail addresses*: snowforesting@163.com (X. Wu), 18919464415@163.com (J. Wang)

## 1 Abstract

Since the impoundment of the Three Gorges Reservoir area in 2003, the potential risks of geological

disasters in the reservoir area have increased significantly, among which the hidden dangers of landslides

are particularly prominent. To reduce casualties and damage, efficient and precise landslide susceptibility

evaluation methods are important. Multiple ensemble models have been used to evaluate the

susceptibility of the upper part of Badong County to landslides. In this study, EasyEnsemble technology

was used to solve the imbalance between landslide and nonlandslide sample data. The extracted

evaluation factors were input into three ensemble models, bagging, boosting, and stacking models, for

training, and landslide susceptibility maps (LSMs) were drawn. According to the importance analysis,

the important factors affecting the occurrence of landslides are altitude, terrain surface texture (TST),

distance to residents, distance to rivers and land use. Comparing the influences of different grid sizes on

the susceptibility results, a larger grid was found to lead to the overfitting of the prediction results.

Therefore, a 30 m grid was selected as the evaluation unit. The accuracy rate, area under the curve (AUC),

recall rate, test set precision, and Kappa coefficient of the multigrained cascade forest (gcForest) model

under the stacking method were 0.958, 0.991, 0.965, 0.946, and 0.91, respectively, which were

significantly better than the values produced by the other two models.

**Keywords:** Landslides · Susceptibility · Ensemble model · Data balance · Three Gorges



## 2 Introduction

As the most common geological disaster, landslides are harmful and destructive and will have a serious impact on human lives and the safety of public facilities. Landslides refer to the disaster phenomenon in which a rock and soil mass on the slope, under the influence of natural conditions and human engineering activities, slides down the slope as a whole or scattered along the failure surface under the action of gravity. At the same time, it also includes the slope mass that is in an unstable state and may evolve into a landslide. Landslide disasters have occurred frequently in the Three Gorges area. The Three Gorges Reservoir project has large potential influences on the environment, geological disasters and the social economy, and the region has received extensive attention. More than 2,500 slope failure sites are known in this area (Skrzypczak et al. 2021; Zou et al. 2021; Chen and Chen 2021); due to the construction of dams, the risk of landslides in the area has increased, and these landslides have huge potential risks. If an effective and accurate landslide susceptibility prediction system can be established, the extent of losses caused by landslide disasters will be minimized (Nsengiyumva and Valentino 2020).

Landslide susceptibility evaluation is particularly important for the prediction and management of landslides. By analysing and calculating the relationship between landslides and landslide influencing factors, landslide-prone areas can be predicted to avoid life and economic losses caused by landslide disasters. This paper evaluated the landslide susceptibility of Badong County using the data balance method and three ensemble model methods of bagging, boosting and stacking.

The occurrence of landslides is related to many environmental factors, and landslide susceptibility assessment explores the connection between them. Through the investigation of historical landslide data, a detailed landslide inventory map was obtained in this paper. Using correlation coefficient analysis,

environmental factors were selected as independent variables. These environmental factors were extracted from digital elevation model (DEM) data, geological maps, Landsat-8 images, basic geographic databases and land cover data. The factors included profile, slope, aspect, altitude, slope length, slope

height, slope pattern, plane curvature, middle slope location, terrain surface texture (TRI), terrain convergence index (TCI), terrain surface convexity (TSC), topographic position index (TPI), TST, valley depth, flow path length, catchment slope, distance to rivers, topographic wetness index (TWI), stream power index (SPI), land use, distance to roads, distance to residents, normalized difference vegetation index (NDVI), and structure data. Using the grid unit as the evaluation unit, the quantitative relationship

between 25 landslide factors and landslide location was calculated by using the representative models of the three ensemble methods of bagging, boosting, and stacking: random forest (RF), extreme gradient boosting (XGBoost) and gcForest. Finally, the evaluation accuracy of landslide susceptibility was verified by comparing the AUC, test set precision, accuracy rate and recall rate with the known landslide.

    In this paper, ArcGIS 10 software, SAGA-GIS software, PyCharm software, and the SPSS 20

statistical program were used for data processing, statistics, and mapping. The technical roadmap of this paper is shown in Figure. 1.





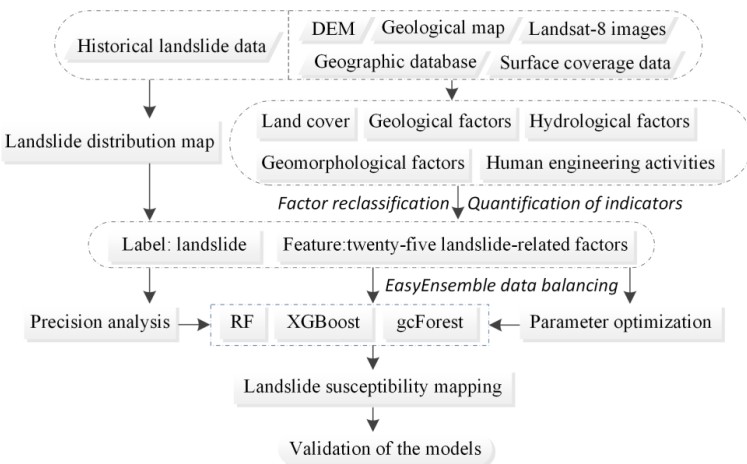

**Fig. 1** Flowchart of this study

## 3 Previous work


Landslide susceptibility is evaluated by determining the combination of factors that have the greatest

impact on the occurrence of landslides after detailed analysis of the landslide's generation conditions and

thus predicting the possibility of landslides in this area (Kayastha and Prabin 2015). Economic

development and the continuous expansion of the scope of human engineering activities have led to the

increasing impact of human beings on the environment, the number of landslide disasters has increased

continually, and the resulting losses are increasingly serious. Therefore, the use of efficient and reliable

landslide hazard evaluation technology for landslide susceptibility evaluation is critical to quickly and

accurately identifying highly prone areas of landslide hazards and predicting the location of new landslide

hazards, which can provide efficient disaster forecasts and reduce losses caused by landslide hazards.

Auxiliary opinions must also be provided for the prevention of geological disasters. To study the mapping

of landslide susceptibility, early researchers proposed various methods and techniques to improve the

accuracy of landslide prediction.

Research on susceptibility evaluation of landslide hazards began in the 1960s. Since the 1990s,



mathematical statistics, probability theory, information theory, and fuzzy mathematics theory have been

continually introduced into the field of geological disaster research. Traditional qualitative research has

gradually moved towards quantitative research—that is, based on data and information—to reflect the

true conditions of landslide geological disasters more objectively and scientifically. At present, GIS-

based methods for landslide geological hazard evaluation can be roughly divided into quantitative

evaluation and qualitative evaluation. With the continuous development of instruments and methods to

obtain spatial data, the quality and quantity of spatial data have also been improved. Data-driven models

have been used in regional LSMs, including support vector machine (SVM) (Yao et al. 2008; Pradhan

2013), RF (Catani et al. 2013; Youssef et al. 2016), artificial neural network (ANN) (Chen et al. 2021;

Gorsevski et al. 2016), and weight-of-evidence (Jayathissa et al. 2019; Hussin et al. 2016) models. In the

data-driven model category, machine learning models have a better prediction effect and higher accuracy

than other approaches, such as expert opinion-based methods and analytic methods (Chowdhuri et al.

2021; Pham et al. 2016). SVM and ANN models are widely used in LSMs and generally can obtain better

prediction results.

Although some machine learning methods perform well in terms of mathematics, explanations of

the internal connection between landslide hazards and various factors remain unavailable. Before

constructing a landslide susceptibility map, to analyse the effect of influencing factors on landslide

occurrence, the mechanism of landslide occurrence must be fully understood, especially in areas

threatened by different types of landslides (Guo et al. 2015). Factor correlation analysis can eliminate the

highly correlated factors influencing landslides, and importance analysis can be used to discern the effect

of factors influencing landslides on landslide occurrence, thereby providing powerful technical means

for selecting important factors influencing landslides and landslide development trend analysis. However,

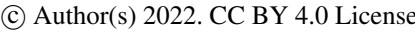



a single learner is prone to underfitting or overfitting. To obtain a learner with high prediction accuracy

and no overfitting, multiple individual learners can be formed into a strong learner through a certain

combination strategy. This method of combining multiple individual learners is called ensemble learning.

The main work of this paper is to compare the prediction effects of three ensemble models, namely,

bagging, boosting, and stacking, on the evaluation of landslide susceptibility in Badong County in the

Three Gorges area. Compared with the above work, the main difference of this research is that the three

ensemble models of bagging, boosting, and stacking were used to model landslide susceptibility, and the

EasyEnsemble method was used to address unbalanced sample data.

## 4 Study area

The Three Gorges area was formed by the severe incision of lower Paleozoic and Mesozoic massive

limestone mountains (Jialinjiang Group, J1) along narrow fault zones in response to Quaternary uplift

(Li et al. 2001). Steep slopes are widely developed on outcrops of erodible or 'soft' materials, and

landslides are common in these areas (Wu et al. 2001). The Three Gorges region of the Yangtze River is

in the mountainous gorge area where Sichuan and Hubei are connected. It contains many mountains and

steep slopes. In the event of heavy rain or earthquakes, disasters such as landslides, mudslides or

rockslides easily occur. The study area is in Badong County (Fig. 2). Located in the middle of Wu Gorge

and Xiling Gorge of the Three Gorges of the Yangtze River, Badong County is the area with the most

complex geological conditions in the region. Folds and faults are widely distributed, and the geological

structure is complex in this area. The whole Badong area has steep terrain with a relative elevation up to

600 m.



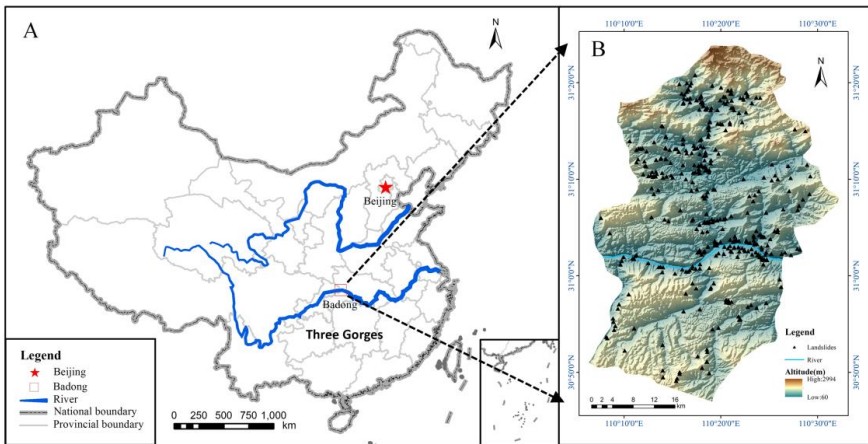

**Fig. 2** Location of the study area in China (A: map of China, B: map of Badong County)

## 5 Data sources

Historical landslide catalogue data include information such as location, geological hazard body,

area, and volume and are used to extract landslide distribution maps. Shuttle Radar Topography Mission

(SRTM)1 DEM data with spatial resolutions of 30 metres are used to extract topography and

geomorphology information. Data acquired from the 1:250,000 national basic geographic database are

used to determine the locations of residential areas, rivers, and roads. Bands 4 and 5 of the 2018 Landsat

8 image are used to obtain the NDVI. The 30-metre global land cover data are the land use data. The

130    NGAC-200,000 national geological map data provide information on the geological structure, strata, and

lithology.

**Table 1** Descriptions of causative factors of landslides

| Data Type | Factors | Source |
|---|---|---|
| Topographic features | Profile | DEM |
| | Slope | DEM |
| | Aspect | DEM |
| | Altitude | DEM |
| | Slope length | DEM |
| | Slope height | DEM |
| | Slope pattern | DEM |
| | Plane curvature | DEM |



| | Middle slope location | DEM |
|---|---|---|
| | TRI | DEM |
| | TST | DEM |
| | TPI | DEM |
| | TSC | DEM |
| | TCI | DEM |
| Hydrological conditions | Valley depth | DEM |
| | Flow path length | DEM |
| | Catchment slope | DEM |
| | Distance to rivers | GIS database |
| | SPI | DEM |
| | TWI | DEM |
| Human engineering activities | Land use | Surface coverage data |
| | Distance to roads | GIS database |
| | Distance to residents | GIS database |
| Surface cover | NDVI | Landsat-8 remote sensing images |
| Basic geology | Structure | Geological map |

## 6 Primary factors of landslide occurrence

135     In this paper, the factors affecting the occurrence of landslides mainly included topography,

geomorphology, hydrological conditions, human engineering activities, surface cover, and basic geology.

ArcGIS software and SAGA-GIS software were used to extract topographic factors from SRTM1 DEM

data, including profile, slope, aspect, altitude, slope length, slope height, slope pattern, plane curvature,

middle slope location, TRI, TST, TPI, TSC, and TCI.

140     SAGA-GIS software was used to extract the valley depth, flow path length, catchment slope,

distance to rivers, SPI, and TWI under hydrological conditions from SRTM1 DEM data. The distance to

rivers, distance to residents, and distance to roads were obtained using the 1:25 million national basic

geographic databases to establish a buffer zone. The NDVI was obtained by calculations of the Landsat-

8 image, the land use type was derived from the 30-metre global land cover data, and the geological

structure was obtained from the geological map data. ArcGIS 10 software was used to extract the

landslide impact factor layer and the landslide layer to the vector points and to make it easy to analyse.



The data set included 2,131,599 rows (number of grids) and 26 columns (25 factors and landslide data).

SPSS 26 statistical software was used to calculate the correlation coefficient analysis for the 25 landslide

impact factors. Most of these 25 factors had low correlation coefficients, and the linear correlation

between these factors was weak. Therefore, 25 landslide impact factors were incorporated into the

landslide susceptibility evaluation system evaluation system to build the probability prediction model of

landslide occurrence.

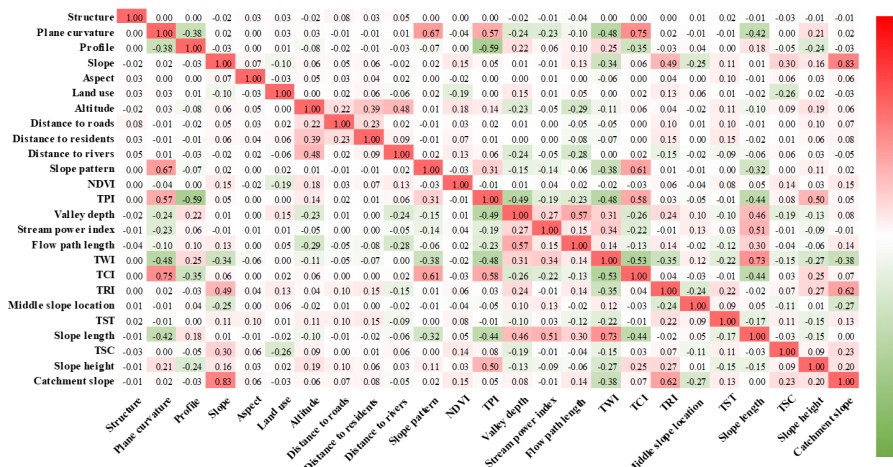

**Fig. 3** Correlation coefficient matrix of the causative factors of landslides

**7 Method for balancing data categories**

The prediction of landslide disasters is a two-class problem in which the prediction results are only

landslides or nonlandslides. An area should contain many more nonlandslide areas than landslide areas.

Assuming that a landslide in the training data belongs to class A and that a nonlandslide belongs to class

B, A: B = 1:99. In this case, if all samples in class A are classified as B, the error rate is only 1/100;

however, if three samples in class B are classified as A, the error rate is 3/100. Achieving higher accuracy

is the objective function of most machine learning algorithms. Such classification algorithms that aim at



maximizing accuracy often ignore the correct classification of small samples such that they often fail to

obtain good prediction results in processing categories with unbalanced samples (Tsai and Lin 2021).

In this case, the algorithm tends to predict all class A samples as class B samples. Landslide disasters

are extremely harmful. High-risk areas are classified as low-risk areas. Once a landslide occurs, it may

cause many casualties and high economic losses. However, if low-risk areas are divided into high-risk

areas, the loss is relatively small (generally, only an economic investment is made to prevent landslide

disasters). The cost of misclassification of the two types of samples is different, and the spatial prediction

of landslide disasters remains a cost-sensitive issue.

The problem of imbalance between the sample categories in landslide areas and nonlandslide areas

can be solved at two levels: the algorithm level and the data level.

At the data level, the following three main data-level solutions are applicable: random sampling,

synthetic minority oversampling technique (SMOTE), and EasyEnsemble technology. For random

sampling, to make the number of samples in the landslide and nonlandslide areas approximately the same,

when selecting the training data set, the same amount of data from landslide and nonlandslide areas are

randomly sampled. The important drawback of this scheme is that if the sample ratio is 1:10 and if

extraction without replacement is used, a maximum of 2 data points can be extracted; that is, a maximum

of 2/11 data points can be used as the training set. This may lead to an insufficient training data set and

make the model training insufficient and unable to achieve the expected prediction accuracy. In addition,

if random sampling with replacement is used, the small sample category is repeatedly sampled many

times, which may cause the model to overfit, resulting in insufficient predictive ability.

The SMOTE algorithm can solve the overfitting problem in random sampling, and its core idea is

to increase the data set of a few categories to achieve the purpose of data equalization (Verbiest et al.




2014). The new sample obtained by this method is not only related to the original sample and its

neighbouring samples but is also different from it. This algorithm can improve the accuracy of landslide

spatial prediction to a certain extent. However, this method is prone to the problem of overlapping

between new samples.

Another problem in random sampling is information loss. This problem can be solved using

EasyEnsemble technology. EasyEnsemble technology trains a number of classifiers for ensemble

learning by repeatedly combining positive samples with the same number of randomly sampled negative

samples. This technology effectively solves the problem of unbalanced data types and reduces the loss

of information for most types of samples caused by undersampling. Therefore, this paper used

EasyEnsemble technology to solve the problem of unbalanced sample types for landslide and

nonlandslide samples. The technical process can be described as follows. (1) The entire training data set

was divided into two categories, namely, majority and minority, which correspond to nonlandslide and

landslide areas, respectively. (2) In each training, the nonlandslide area was randomly divided into n

parts, and all samples in the landslide area were 1 part. (3) One piece was randomly selected from the

nonlandslide sample to form a new training data subset, together with the landslide area. This subset was

used to train the classifier to obtain the classification result and save it. (4) Steps (2) and (3) were repeated

n times to obtain n classification results. (5) The average of the category scores of the n classification

results was calculated to obtain the final classification result.

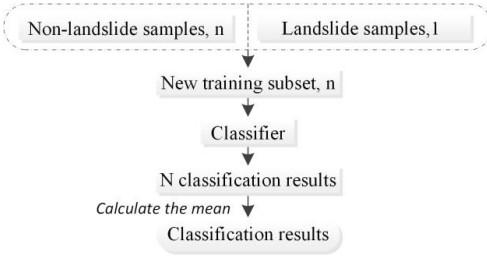





**Fig. 4** EasyEnsemble technology solves the problem of unbalanced landslide samples

The solution at the algorithm level is mainly to use the cost matrix to set the weights corresponding

to different categories. The idea is that the cost of misclassification of different categories is different,

and different categories are assigned different penalty coefficients in the algorithm. The purpose is to

distinguish as few samples as possible.

## 8 Ensemble model

Landslide susceptibility is evaluated by predicting the possibility of landslides in a certain area by

setting the most favourable combination of factors for landslide occurrence after analysing the landslide

occurrence conditions. Many scholars have used landslide susceptibility evaluations to find potential

high-risk areas within a region to reduce the dangers of landslides, and they have obtained good results.

The content of landslide susceptibility evaluation includes the division of evaluation units and the

selection of evaluation factors. Choosing a suitable model can obtain better prediction results for

landslide susceptibility evaluation.

In 1962, the idea of ensemble learning began to appear. The first appearance of a cascading

multiclassifier ensemble system was in the book by Sebestyen. Ensemble learning entered researchers'

field of vision in the 1990s when Hansen et al. proposed a neural network ensemble model that used

voting to integrate output results to obtain a better classifier than a single neural network. Bagging,

boosting and stacking are three typical paradigms of ensemble learning. By combining several machine

learning algorithms into a meta-algorithm of a prediction model, the effect of reducing errors or

extracting predictions can be achieved.

The bagging ensemble algorithm (Suzuki and Ohkura 2016) is an ensemble learning algorithm in

the field of machine learning originally proposed by Leo Breiman. The combination of the bagging



integration algorithm and other algorithms can effectively enhance the prediction accuracy and stability

of classification methods. The main content of the algorithm involves taking a training set S of size N

and evenly selecting n subsets $S_i$ of size N from S with replacement (self-service sampling method) as a

new training subset. By using these n training subsets, n training results can be obtained, and the analysis

results can be obtained through strategies such as averaging or voting. The main advantage is that it can

generate formation learners that are not dependent on each other in parallel. The bagging ensemble

algorithm is suitable for the prediction of small sample data sets and has a good application effect in the

field of machine learning.

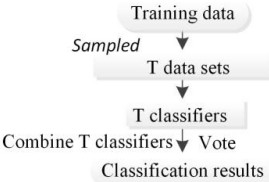

**Fig. 5** Flowchart of the bagging method

The boosting algorithm (A et al. 2002) first uses the training set and initial weight to train weak

learner 1. Weak learner refers to a learner whose generalization performance is slightly better than

random guess. Usually, different weights are given according to their classification accuracy, and the

samples with low accuracy are given higher weights. The samples with higher weights are considered in

the subsequent learners. Then, weak learner 2 is trained according to the training samples after adjusting

the weights. Repeat the above steps t times to generate T base classifiers. The boosting framework

algorithm weights and fuses the N base classifiers to produce a better result classifier. After weighted

fusion of weak learners, the data will usually be reweighted to strengthen the classification of previously

classified wrong data points. In the training of the boosting algorithm, the classifier is trained based on

the samples with errors in the previous classification such that the algorithm can reduce the classification





error rate of the model; however, as the training progresses, the entire model classifies the training set

correctly as the rate continues to increase, and the variance of the model increases. However, random

sampling of features for training can reduce the correlation between models, thereby reducing the overall

variance of the model (Benmokhtar and Huet 2006; Gou et al. 2019; Liang et al. 2021; Woniak et al.

2014).

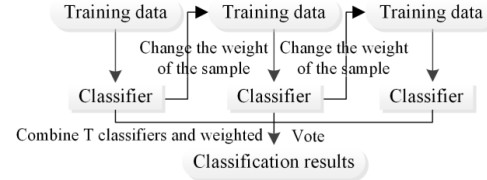


**Fig. 6** Flowchart of the boosting method

The stacking method (Rahman et al. 2020) first trains the primary learner, and then the prediction

result of the primary learner is used as the new input to train the secondary learner. In the training phase,

the secondary learner is generated by the primary learner. If the prediction results of the primary learner

are directly used to generate the training set of the secondary learner, the risk of overfitting is high.

Therefore, the initial training set is divided into k parts, and cross-validation is used to train each learner

(Xia et al. 2020).

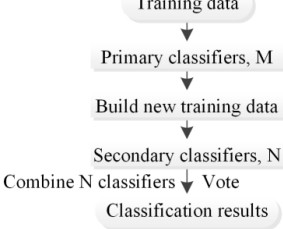

**Fig. 7** Flowchart of the stacking method

Each of the three ensemble methods, bagging, boosting, and stacking, has multiple models. This

article uses three representative models in the three ensembles as the landslide susceptibility prediction




model: the RF in the bagging model, the XGBoost model in the boosting model, and the gcForest model

in the stacking model.

The random forest model is a classifier containing multiple decision trees whose basic unit is a

decision tree, each of which is a classifier (Asadi et al. 2021). Random forest focuses on the ensemble

learning of decision trees. After the decision tree is integrated, the model uses voting to determine the

prediction result; that is, the prediction result is the category with the most votes. The random forest

model is suitable for large-scale data prediction, but other models obtain poor prediction results because

of the high dimensionality of the sample. The accuracy of the random forest model for most learning and

prediction tasks can reach the same level as other models, and little overfitting occurs. In the process of

competition and practical application, the random forest model is widely used. The model has two

important parameters, including the number of subtrees and the maximum number of features allowed

for a single decision tree.

The XGBoost algorithm is an improvement method. The core idea of the improvement algorithm is

that multiple experts individually judge a complex task and then perform a proper synthesis to reach the

conclusion. The conclusion drawn after the synthesis is better than any one of the experts alone. The

XGBoost algorithm is based on the regression tree model. The basic idea is to repeatedly extract some

variables to construct the regression tree model, obtain hundreds of regression tree models, and combine

them linearly to obtain the final model.

The gcForest integration method is a new method based on decision tree forest aggregation. The

gcForest integration method can make the data set of gcForest automatically learn its representation

structure. The reason is that the method can automatically generate a decision tree forest with a higher-

dimensional cascade structure. For example, when the decision tree input has a higher-dimensional data





set, the gcForest method can use a multigranular scanning method to increase the dimensional features

such that gcForest can express the awareness of structural learning. In addition, the gcForest method can

automatically set its model complexity and adaptively determine the number of layers of cascading

forests, making it more capable of training data of different sizes. In other words, gcForest automatically

stops the next calculation when the calculation result of the last cascade layer is lower than the expected

value. Therefore, the gcForest method is suitable for both small-scale data and large-scale data training.

In terms of the number of model parameters, the gcForest model has less than the ANN model, and it is

also reliable for the parameter setting of the neural network with fewer settings.

## 9 Landslide susceptibility mapping

The grid unit was used as the evaluation unit of this study, and the multivalue extraction-to-point

function in ArcGIS 10 software was used to extract 25 factors that influence landslides. The data set of

the study area had 2,131,599 rows (number of grids) and 26 columns (25 factors and landslide data).

Under the 30 m grid, 269,421 pieces of data were labelled landslides, and 2,104,657 pieces of data were

labelled nonlandslides; the ratio of landslide data to nonlandslide data was approximately 1:10. Therefore,

25,000 pieces of landslide data and 205,000 pieces of nonlandslide data were randomly selected, 5,000

pieces of landslide data were removed, 5,000 pieces of nonlandslide data were taken as test data, and the

remaining 20,000 pieces of landslide data and 200,000 pieces of nonlandslide data were used as training

data. Extracting training data in this way can make the ratio of landslides and nonlandslides in the training

data close to the actual situation in the study area. Because the impact of grids of different sizes on

landslide susceptibility needs to be compared in the future, the data of 60 m and 90 m grids were

processed similarly and organized into training sets and test sets. After the EasyEnsemble data balance

was performed on the data set, the data set was used to train the RF model in the bagging algorithm, the



XGBoost model in the boosting algorithm, and the gcForest model in the stacking algorithm, and the

train results were used to predict the probability of landslides for all samples from each model. The

prediction results were added to the attribute table of the vector points in the study area, and then the

vector point data were converted into raster data to draw the landslide prone area map of the three models.

Feature importance measures the contribution of each input feature to the prediction results of the

model, which highlights the degree of correlation between the feature and the target. This paper

calculated the importance of 25 factors for the three tested models. The test results show that the altitude,

TST, distance to residents, distance to rivers and land use are the main factors that affect landslide

susceptibility.

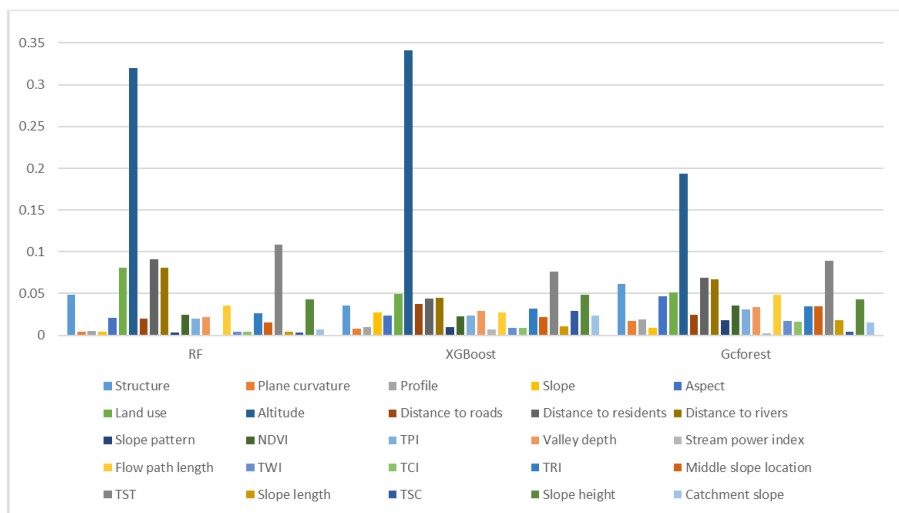


**Fig. 8** Feature importance measures (FIMs) of causative factors of landslides in different models

      The influence of altitude on a landslide distribution is mainly reflected in the local water collection

platform caused by the topographic slope differences between different altitude ranges, the differences

between the intensities of free surface and human engineering activities that are prone to landslides in

different altitude ranges, and the characteristics of different vegetation types, coverages and atmospheric



rainfall levels in different altitude ranges. Therefore, height is an important factor in landslide disaster-prone environments. Based on an analysis of the statistical zoning table of landslides combined with elevation, the frequency ratio is greater than 1 in the altitude range from 49 m to 594 m. With increasing elevation, the frequency ratio decreases, which shows that landslides are mainly distributed in the low

elevation range.

Terrain surface texture is one of the main parameters for representing the development characteristics of landforms. In places with complex terrain, such as ridges and valleys, the texture feature values are large, while in smooth and flat places, the texture values are small. According to the statistical table of landslide zoning based on terrain surface textures, the frequency ratio of terrain surface texture values is greater than 1 in the range of 0.06 to 14.31, and the frequency ratio is largest in the range of

values is greater than 1 in the range of 0.06 to 14.31, and the frequency ratio is largest in the range of 0.06 to 9.03, indicating that the landslides in the study area are mostly distributed in areas with relatively smooth and flat terrain.

Human engineering activities are human engineering construction activities related to resource exploitation and infrastructure construction processes that use certain engineering and technical means,

including planning, design, construction, mining and operation. Human engineering activities can cause land erosion and change the original landform. Such activities cause gradual and great harm. The areas where human engineering activities occur are often located near residential areas (examples include urban construction, irrigation activities, and traffic construction); thus, the distance from residential areas was taken as an evaluation factor. According to the statistical table of landslide zoning based on the

distance from the residential area, the frequency ratio between 0 and 1040 m from the residential area is greater than 1, and the maximum ratio occurs within 614 m, indicating that the closer to the residential area the activity is, the more landslide disasters are likely to occur.



The impact of the river on the landslide disaster in the study area is mainly manifested in the lateral erosion and erosion-based cutting of the river water on the river valley bank slope. On the one hand, the

river continuously cuts down to make the bank slope higher and steeper; on the other hand, it continuously washes the slope toe, causing the slope to always be in an unstable state. It is one of the important factors for the formation of new landslide masses and the revival of old landslide masses. Therefore, this paper selected the distance from the river as an evaluation factor to consider the impact of rivers on landslide disasters. The zoning statistics of the landslides based on the distance to the river

metric indicate that the frequency ratio is greater than 1 within the range of 451.15 m from the river, and the frequency ratio decreases with increasing distance from the river, indicating that landslides are more likely to occur in areas that are closer to the river.

Land use refers to the long-term or periodic use, protection and transformation of land by using certain transformation means based on the natural attributes and characteristics of the land of interest.

Five main types of land use are employed in the study area, including cultivated land, forest, grassland, water bodies and artificial surfaces. According to the statistical zoning table of land use for landslides, the regional frequency ratios of artificial surfaces, cultivated land and water bodies are greater than 1 (especially the frequency ratio of artificial surfaces, which is the highest), while the frequency values of grassland and forests are less than 1. This shows that the landslides in the study area are more distributed

in the areas where artificial surfaces, cultivated land and water bodies are located, and few landslides are contained in forests and grasslands.

**Table 2** Statistical zoning table for the top five impact factors

| The evaluation factors | Classification level | Number of pixels in domain | Number of landslides | Percentage of domain (%) | Percentage of landslides (%) | FR |
|---|---|---|---|---|---|---|
| | 49-594 | 533,942 | 18,846 | 0.25 | 0.70 | 2.79 |
| Altitude | 594-937 | 579,737 | 6,980 | 0.27 | 0.26 | 0.95 |
| | 937-1,241 | 420,759 | 802 | 0.20 | 0.03 | 0.15 |



| | | | | | |
|---|---|---|---|---|---|
| | 1,241-1,561 | 349,147 | 333 | 0.16 | 0.01 | 0.08 |
| | 1,561-1,984 | 179,724 | 11 | 0.08 | 0.00 | 0.00 |
| | 1,984-3,096 | 67,119 | 0 | 0.03 | 0.00 | 0.00 |
| Terrain surface texture level | 0.06-9.03 | 287,378 | 9,355 | 0.13 | 0.35 | 2.57 |
| | 9.03-14.31 | 456,151 | 8,470 | 0.21 | 0.31 | 1.47 |
| | 14.31-18.88 | 500,238 | 5,491 | 0.23 | 0.20 | 0.87 |
| | 18.88-23.28 | 433,011 | 2,840 | 0.20 | 0.11 | 0.52 |
| | 23.28-28.21 | 313,383 | 659 | 0.15 | 0.02 | 0.17 |
| | 28.21-44.91 | 140,267 | 127 | 0.07 | 0.00 | 0.07 |
| Distance to residents (m) | 0-614.82 | 625,674 | 13,253 | 0.29 | 0.49 | 1.67 |
| | 614.82-1,040.46 | 773,592 | 11,579 | 0.36 | 0.43 | 1.18 |
| | 1,040.46-1,489.75 | 473,118 | 1,977 | 0.22 | 0.07 | 0.33 |
| | 1,489.745-2,104.56 | 174,421 | 133 | 0.08 | 0.00 | 0.06 |
| | 2,104.56-3,121.37 | 60,809 | 0 | 0.03 | 0.00 | 0.00 |
| | 3,121.37-6,029.93 | 22,547 | 0 | 0.01 | 0.00 | 0.00 |
| Distance to rivers (m) | 0-451.15 | 820,030 | 17,434 | 0.38 | 0.65 | 1.68 |
| | 451.15-1,008.46 | 593,174 | 6,963 | 0.28 | 0.26 | 0.93 |
| | 1,008.46-1,645.39 | 412,379 | 1,909 | 0.19 | 0.07 | 0.37 |
| | 1,645.39-2,415.00 | 207,723 | 547 | 0.10 | 0.02 | 0.21 |
| | 2,415.00-3,529.62 | 75,110 | 89 | 0.04 | 0.00 | 0.09 |
| | 3,529.62-6,767.31 | 22,012 | 0 | 0.01 | 0.00 | 0.00 |
| Land use | Cultivated land | 580,187 | 16,364 | 0.27 | 0.61 | 2.23 |
| | Forest | 1,414,552 | 8,388 | 0.66 | 0.31 | 0.47 |
| | Grassland | 93,621 | 813 | 0.04 | 0.03 | 0.69 |
| | Water bodies | 32,002 | 768 | 0.02 | 0.03 | 1.90 |
| | Artificial surfaces | 8,823 | 605 | 0.00 | 0.02 | 5.42 |

According to the landslide occurrence possibility predicted by the model, the landslide susceptibility

zoning map is drawn. The study area has five types of susceptibility levels: very low, low, medium, high,

and very high. The RF model is visible on the susceptibility map. Compared with other models, more

places are divided into extremely high landslide-prone areas and high landslide-prone areas. The gcForest

model predicts the least amount of extremely high and high landslide-prone areas. Most of the very low

landslide-prone areas are in the south and north of the study area. The extremely high landslide-prone





areas and high landslide-prone areas ascertained by the three models are basically located along the

Yangtze River and in the middle and upper sections of the study area. The Rf model predicts many areas

that have not experienced landslides in the past as areas with higher susceptibility, such as the north bank

in the western section of the Yangtze River in the study area. The XGBoost model basically predicts the

locations where landslides have occurred as areas with higher susceptibility, and the gcForest model

predicts very few areas as areas with higher susceptibility, but most of the locations where landslides

have occurred are predicted to be more landslide-prone areas.

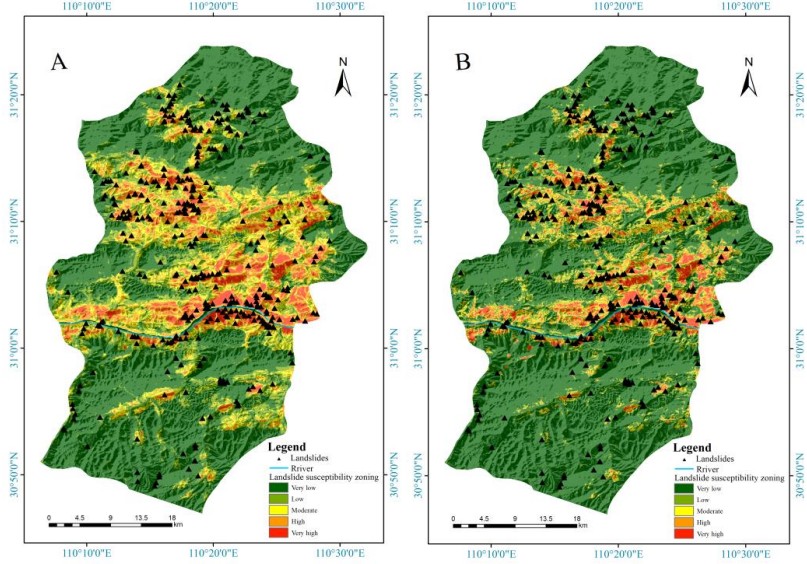





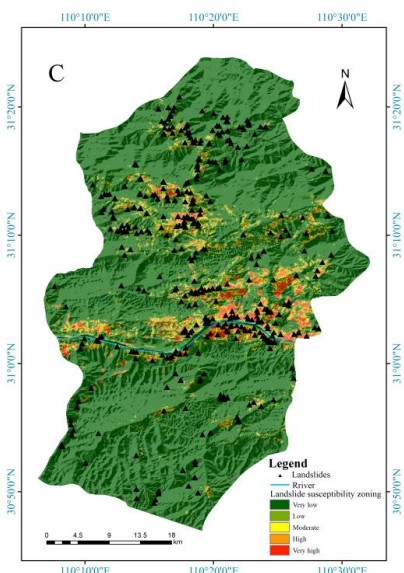

**Fig. 9** Landslide susceptibility zoning maps produced by three ensemble models: (A) RF model, (B) XGBoost
model, and (C) gcForest model

Through the susceptibility results yielded by the three models under different grid sizes and the

statistical zoning table of landslides, the frequency ratio under each susceptibility level was calculated.

The table shows that the higher the susceptibility is, the higher the frequency ratio is, indicating that most

landslides are in the highly and extremely highly prone areas classified by the model, and the prediction

results are reasonable. In the case in which the same model is used, taking the gcForest model as an

example, as the grid increases from 30 m to 90 m, the grid proportion for landslides in extremely highly

prone areas changes little, from 97% to 88%; meanwhile, the grid proportion for highly prone areas

increases by more than three times, from 3% to 10%. Therefore, the frequency ratio of highly prone areas

also decreases from 30.1380 to 8.9695, and the prediction effect of the model worsens. Therefore, a

smaller grid should be selected as the evaluation unit for the study of landslide susceptibility. When the

grid size remains the same, the frequency ratio of the extremely highly prone area of the RF model is the

lowest, and the frequency ratio of the gcForest model is the highest, indicating that the gcForest model





predicts a smaller extremely highly prone area but contains more landslides; thus, its prediction effect is

the best.


**Table 3** RF zoning model of landslide susceptibility

| Grid size | Landslide susceptibility level | Number of pixels in domain | Number of landslides | Percentage of domain (%) | Percentage of landslides (%) | FR |
|---|---|---|---|---|---|---|
| 30 m | Very low | 1,153,876 | 38 | 0.54 | 0.00 | 0.0026 |
| | Low | 347,494 | 382 | 0.16 | 0.01 | 0.0869 |
| | Moderate | 260,594 | 2245 | 0.12 | 0.08 | 0.6812 |
| | High | 239,440 | 7737 | 0.11 | 0.29 | 2.5551 |
| | Very high | 128,884 | 16539 | 0.06 | 0.61 | 10.1469 |
| 60 m | Very low | 219,759 | 5 | 0.41 | 0.00 | 0.0018 |
| | Low | 110,687 | 56 | 0.21 | 0.01 | 0.0400 |
| | Moderate | 77,065 | 342 | 0.15 | 0.05 | 0.3505 |
| | High | 70,236 | 1192 | 0.13 | 0.18 | 1.3403 |
| | Very high | 52,706 | 5122 | 0.10 | 0.76 | 7.6745 |
| 90 m | Very low | 94,847 | 10 | 0.40 | 0.00 | 0.0083 |
| | Low | 48,629 | 44 | 0.20 | 0.01 | 0.0709 |
| | Moderate | 36,009 | 124 | 0.15 | 0.04 | 0.2700 |
| | High | 32,648 | 501 | 0.14 | 0.17 | 1.2031 |
| | Very high | 25,509 | 2352 | 0.11 | 0.78 | 7.2290 |

**Table 4** XGBoost zoning model of landslide susceptibility

| Grid size | Landslide susceptibility level | Number of pixels in domain | Number of landslides | Percentage of domain (%) | Percentage of landslides (%) | FR |
|---|---|---|---|---|---|---|
| 30 m | Very low | 1,544,816 | 80 | 0.73 | 0.00 | 0.0041 |
| | Low | 196,104 | 272 | 0.09 | 0.01 | 0.1097 |
| | Moderate | 145,685 | 1082 | 0.07 | 0.04 | 0.5872 |
| | High | 135,596 | 4738 | 0.06 | 0.18 | 2.7628 |
| | Very high | 108,089 | 20770 | 0.05 | 0.77 | 15.1937 |
| 60 m | Very low | 318,751 | 18 | 0.60 | 0.00 | 0.0045 |
| | Low | 71,661 | 52 | 0.14 | 0.01 | 0.0573 |
| | Moderate | 49,886 | 175 | 0.09 | 0.03 | 0.2770 |
| | High | 44,705 | 652 | 0.08 | 0.10 | 1.1516 |
| | Very high | 45,451 | 5821 | 0.09 | 0.87 | 10.1126 |
| 90 m | Very low | 138,775 | 28 | 0.58 | 0.01 | 0.0158 |
| | Low | 32,145 | 61 | 0.14 | 0.02 | 0.1487 |
| | Moderate | 22,725 | 94 | 0.10 | 0.03 | 0.3242 |
| | High | 21,175 | 266 | 0.09 | 0.09 | 0.9846 |
| | Very high | 22,823 | 2583 | 0.10 | 0.85 | 8.8705 |

**Table 5** gcForest zoning model of landslide susceptibility

| Grid size | Landslide susceptibility level | Number of pixels in domain | Number of landslides | Percentage of domain (%) | Percentage of landslides (%) | FR |
|---|---|---|---|---|---|---|





| | | | | | | |
|---|---|---|---|---|---|---|
| | Very low | 1,842,089 | 62 | 0.86 | 0.00 | 0.0027 |
| | Low | 105,676 | 104 | 0.05 | 0.00 | 0.0778 |
| 30 m | Moderate | 65,196 | 185 | 0.03 | 0.01 | 0.2244 |
| | High | 50,857 | 414 | 0.02 | 0.02 | 0.6437 |
| | Very high | 66,472 | 26,177 | 0.03 | 0.97 | 31.1380 |
| | Very low | 328,806 | 24 | 0.62 | 0.00 | 0.0058 |
| | Low | 72,528 | 59 | 0.14 | 0.01 | 0.0642 |
| 60 m | Moderate | 49,390 | 132 | 0.09 | 0.02 | 0.2110 |
| | High | 40,401 | 385 | 0.08 | 0.06 | 0.7524 |
| | Very high | 39,329 | 6,118 | 0.07 | 0.91 | 12.2830 |
| | Very low | 118,010 | 16 | 0.50 | 0.01 | 0.0106 |
| | Low | 42,356 | 42 | 0.18 | 0.01 | 0.0777 |
| 90 m | Moderate | 28,854 | 92 | 0.12 | 0.03 | 0.2499 |
| | High | 25,179 | 222 | 0.11 | 0.07 | 0.6911 |
| | Very high | 23,244 | 2,660 | 0.10 | 0.88 | 8.9695 |

## 10 Validation of the models

In an experiment comparing the influences of different grid sizes on the susceptibility results, the receiver operating characteristic (ROC) curves and AUC values of each model under different grid sizes were obtained. The ROC curves and AUC values were calculated by using the probability obtained from the data predicted by the three models. The numbers of grids with different sizes are different. The number of grids with a 30 m grid size is 2,131,599, including 26,942 landslide grids. Under a grid size of 60 m, the number of grids is 532,335, including 6715 landslide grids. The number of grids under a 90 m grid is 238,296, including 3,009 landslide grids. Comparing different grid sizes under the same model, the AUC value was found to decrease with increasing grid size. The AUC value was largest under the 30-metre grid, and the AUC value was smallest under the 90 m grid. Comparing different models with the same grid size, the AUC value of the gcForest model was highest and that of the RF model was lowest, indicating that the prediction effect of the gcForest model is the best.





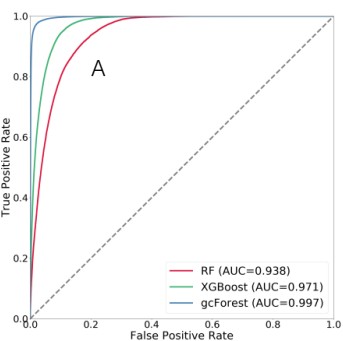

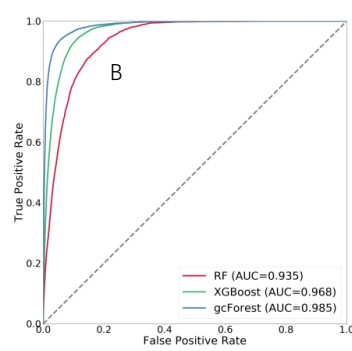

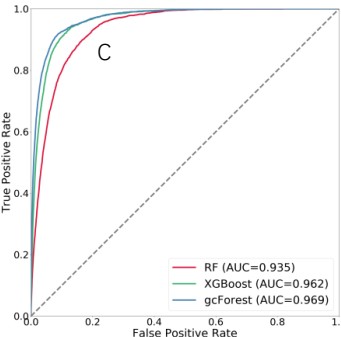


**Fig. 10** ROC curves of all data under different grid sizes (A: 30 m, B: 60 m, and C: 90 m)

Comparing the effects of different grid cell sizes on the susceptibility results, the larger the grid cell

is, the higher the accuracy of the training data and the lower the accuracy of the test data. This proves

that model overfitting occurs with increasing mesh size. As the grid becomes larger, the gap between the

accuracies of the training data and test data becomes larger, especially for the gcForest model. When the

grid size is 90 m, the difference between the training data and test data accuracies of the gcForest model

is as high as 15.2%. Therefore, in this paper, a 30 m grid was selected as the evaluation unit for landslide

susceptibility modelling such that better prediction accuracy could be obtained without overfitting.

**Table 6** Accuracies of the training data and test data under different grid sizes

| model | | 30 m | 60 m | 90 m |
|---|---|---|---|---|
| rf | train | 0.873 | 0.890 | 0.912 |
| | test | 0.862 | 0.851 | 0.838 |
| XGBoost | train | 0.929 | 0.960 | 0.988 |





| | | | | |
|---|---|---|---|---|
| | test | 0.912 | 0.887 | 0.872 |
| gcForest | train | 0.999 | 0.999 | 0.999 |
| | test | 0.958 | 0.890 | 0.847 |


The following table indicates the prediction accuracy of the RF, XGBoost and gcForest models for samples in the study area. The AUC is an evaluation index used to measure the advantages and disadvantages of binary classification models. From the definition, the AUC can be obtained by summing the areas of each part under the ROC curve. Its value represents the probability that the predicted positive

case is ahead of the negative case. The recall rate indicates how many positive examples in the sample are predicted correctly. The accuracy is the number of samples that predict the correct prediction of the positive class, which accounts for the proportion of the number of all positive samples predicted. The kappa coefficient can be used to test the consistency and evaluate the accuracy of a multiclass classification model. Whether the actual classification results of the model are consistent with the

prediction results is the consistency of the classification problem. The kappa coefficient is obtained by calculating the confusion matrix, and its value is between -1 and 1, which is generally greater than 0. The accuracy rate, AUC value, recall rate, test set precision, and Kappa coefficient of the gcForest model in the stacking method are 0.958, 0.991, 0.965, 0.946, and 0.91, respectively, which are significantly better than the values of the other two models.

**Table 7** Statistical measures of different methods obtained on the training and test sets

| Data set | Learning methods | Performance | | | | |
|---|---|---|---|---|---|---|
| | | Accuracy | AUC | Recall | Precision | Kappa |
| Training set | RF | 0.873 | 0.943 | 0.933 | 0.808 | 0.749 |
| | XGBoost | 0.929 | 0.979 | 0.97 | 0.89 | 0.861 |
| | gcForest | 0.999 | 0.999 | 0.999 | 0.999 | 0.999 |
| Test set | RF | 0.862 | 0.932 | 0.914 | 0.805 | 0.725 |
| | XGBoost | 0.912 | 0.968 | 0.955 | 0.875 | 0.819 |
| | gcForest | 0.958 | 0.991 | 0.965 | 0.946 | 0.91 |



## 11 Discussion and conclusions

This paper is a comparative study of multiple ensemble models of landslide susceptibility assessment in the upper half of Badong County of the Three Gorges area. The landslide data were

obtained from historical landslide records. In this landslide susceptibility analysis, 25 factors influencing landslides, including slope, aspect, plane curvature, profile curvature, and elevation, were used. According to the importance analysis, the important factors affecting the occurrence of landslides are the altitude, TST, distance to residents, distance to rivers and land use. Comparing the influences of different grid sizes on the susceptibility results, larger grids lead to the overfitting of the prediction results.

Therefore, a 30 m grid was selected as the evaluation unit, and the study area contains 2,131,599 grid units. Due to the imbalance between the sample landslide data and the nonlandslide data, ensemble data balance processing was performed on the sample to construct the test data and the training data. Using the RF model in the bagging model, the XGBoost model in the boosting model, and the gcForest model in the stacking model for training and prediction, a landslide susceptibility map was generated. According

to the landslide susceptibility map, the locations of the extremely high landslide-prone areas and high landslide-prone areas in the three models are basically consistent with the locations of historical landslides. The surrounding areas of the Yangtze River and its tributaries and the middle and upper areas of the study area are very prone to landslides.

The landslide susceptibility map was verified using the success rate curve to compare with known

landslides. The quantitative results show that the order of the AUC values from small to large are the RF model, the XGBoost model, and the gcForest model. The accuracy rate, AUC value, recall rate, test set precision, and Kappa coefficient of the gcForest model in the stacking method are 0.958, 0.991, 0.965, 0.946, and 0.91, respectively, which are significantly better than the values of the other two models.



**CRediT authorship contribution statement**

**Xueling Wu:** Conceptualization, Methodology, Visualization, Writing – review & editing, Data curation,

Project administration. **Junyang Wang:** Writing – original draft, Software, Methodology, Visualization, Validation.

**Declaration of competing interests**

The authors declare that they have no known competing financial interests or personal relationships that could

have appeared to influence the work reported in this paper.

**Acknowledgements**

This study was jointly supported by the National Natural Science Foundation of China (42071429; 41871355).

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
