# Peer review of "Application of bagging, boosting and stacking ensemble and EasyEnsemble methods to landslide susceptibility mapping in the Three Gorges Reservoir area of China"

_EGUsphere, 2022_

## Author Comment (AC1)

**To Reviewer #1**

*General comments: This manuscript discussed the application of several machine learning models in landslide susceptibility analysis. However, I don't think it is worth being published in a high quality journal like NHESS. Here you can find my concerns:*

**Response:** We thank you for your recommendation and valuable comments, which have ultimately improved this manuscript. We greatly appreciate your extensive and thoughtful review of our manuscript. According to your comments, we have made point-by-point corrections, which we hope will meet your approval.

**Point-by-point responses to your detailed comments**:

*1.Comment: My biggest concern is from the novelty of this study. What is the new thing of it? A quick Google search showed that too many similar researches have been published. Most of them are characterized by the key words like "machine learning", "landslide (or other hazards) susceptibility". And the most important objective of such studies is to compare the ability of different models. But in my opinion, it doesn't make sense when you compare too many models. They are just regular exercises on this topic.*

**Response:** Thank you for your careful insights. We need to explain the innovation of this paper. Generally, for the whole study area, the landslide area accounts for a small percentage of the total, and the nonlandslide area

accounts for the majority. If the data are not balanced, the algorithm prefers to predict a small number of landslide areas as nonlandslide areas to achieve improved accuracy. Landslides cause great harm. The high-risk areas of landslides are wrongly categorized as low-risk areas of landslides. Once a landslide occurs, it may cause casualties and economic losses. Therefore, this paper uses the EasyEnsemble method to solve the landslide data imbalance problem. It is true that studies of integrated models in terms of landslide susceptibility are not uncommon. Some studies (Hu et al. 2020; Zhang et al. 2022) have applied integrated models to landslide susceptibility modelling, but few articles have compared and analysed three integrated models with respect to landslide susceptibility. Some landslide susceptibility studies (Zheng 2020) have used a variety of integrated models but did not consider the problem of landslide data imbalance or different study areas. The innovation of this paper is that it can apply a variety of integrated models in combination with the EasyEnsemble data balancing method to the Three Gorges area of China.

Hu X, Zhang H, Mei H, Xiao D, Li M (2020) Landslide susceptibility mapping using the stacking ensemble machine learning method in lushui, southwest china. Applied Sciences, 10(11), 4016.

Zheng, H (2020) Improved landslide assessment using support vector machine with bagging, boosting, and stacking ensemble machine learning framework in a mountainous watershed, Japan. Landslides, 17(3), 641-658.

Zhang T, Fu Q, Wang H, Liu F, Han L (2022) Bagging-based machine learning algorithms for landslide susceptibility modeling. Natural Hazards, 110(2), 823-846.

**2.Comment**: *The structure of the MS is confusing. It is not using a widely accepted template for paper: Introduction—Methods—Study area—Results—Discussion—conclusion.*

**Response:** Thank you for pointing this out. The content of this paper includes the introduction, methods, research areas, results, discussions, and conclusions. Previous work corresponds to the research field section; Primary factors of landslide occurrence, Method for balancing data categories, and Ensemble model correspond to the method section; Landslide susceptibility mapping, Validation of the models corresponds to the results and discussion sections, and the discussion and conclusion titles should be change to conclusion. Sections of the current dissertation can be added to this widely accepted dissertation template as secondary headings: Introduction - Methods - Research Areas - Results - Discussion - Conclusions.

**3.Comment**: *It seems that a real Discussion section is missing.*

**Response:** Thank you for your thoughtful insights. The Landslide susceptibility mapping and Validation of the models sections of this paper contain the results and discussion sections, in which the structure of the paper is to present the results and discuss and analyze them. For example, in the

Landslide susceptibility mapping section, the results of the landslide susceptibility zoning map are displayed and the distribution of susceptibility zoning is discussed and analyzed.

**4.Comment**: *You selected 25 factors as input data of the model, but why are these factors not others? I mean all these factors are from literature and experience, aren't they? How do you justify they are necessary, and the factors not selected by you are not necessary?*

> **Response:** Thank you for your careful insights. The paper selects these factors by reading literatures about landslide susceptibility, taking some commonly used factors as evaluation factors in this paper, and then screening evaluation factors through SPSS software collinearity analysis to remove evaluation factors with high correlation.

**5.Comment**: *In Abstract and Conclusion, quantitative results are really few.*

> **Response:** Thank you for your careful insights. Many quantitative results have been presented and analyzed in the previous sections, and are not re-introduced in the final conclusions, only summarizing the main issues.
>
> ***Thank you very much for your insightful and detailed comments.***

---

## Author Comment (AC2)

**To Reviewer #2**

*General comments: The authors presented a comparison of bagging, boosting and stacking ensemble methods to evaluate the landslide susceptibility mapping in the Three Gorges Reservoir area of China. Although the manuscript is well written, but the presented methods and presentation of data alacks novelty. One can easily guess that which models work better in the start without further reading the contents, as these has been done in several previous studies. From a readers viewpoint, I would like to see real discussion of science; however which is missing in this paper.*

**Response:** We thank you for your recommendation and valuable comments, which have ultimately improved this manuscript. We greatly appreciate your extensive and thoughtful review of our manuscript. According to your comments, we have made point-by-point corrections, which we hope will meet your approval.

**Point-by-point responses to your detailed comments**:

*1.Comment: For example, the authors analysed 25 factors (and surprisingly to me this do not find a collinearity problem); how these factors contribute to the landslides in TGD area?. I also see that feature importance is signifcantly high for Altitude. Some other important factors do not contribute at to the model as well such as the slope. Explanation of that adds to discussion.*

**Response:** Thank you for your careful insights. Through the feature importance analysis, the paper concludes that altitude, terrain surface texture (TST), distance to residents, distance to rivers and land use are five important factors that affect the occurrence of landslides. Statistical landslides are mainly distributed in which range of each factor and how the factor causes the landslide to occur. Due to the large number of evaluation factors, only the evaluation factors that have a greater impact on the occurrence of landslides are analyzed.

**2.Comment**: *Also, authors have computed the results of zoning in Table 3,4 and 5. What is their meaning to a reader ?*

**Response:** Thank you for your careful insights. For readers, from Tables 3, 4, and 5, it can be seen that the number of grids, the number of landslide grids, and the frequency ratio of the three models at each susceptibility zoning level at different grid sizes. A high frequency ratio means that more landslide grids are divided under this susceptibility level, and a high frequency ratio of the extremely high-prone areas and high-prone areas indicates that the model predicts better.

**3.Comment**: *Additionally, authors hsould validate the model not in the training site. They should have choose the adjoining catachments to check whether the result still hold valid (Like AUC of 0.95).*

**Response:** We thank you for your valuable comments. The model accuracy rate in Table 6 includes the accuracy rate of the training data and the accuracy rate of the test data, and the test data does not participate in the model training.

*4.Comment*: *Again, the comparisojn of 30-60-90 m grid size is inappropriate, as these are again known from several past works.*

**Response:** Thank you for your careful insights. It is impossible to know which evaluation unit is suitable for landslide susceptibility evaluation in this area without comparing the evaluation units of different sizes. After comparing the 30-60-90 m grids, we found that with the increase of the grid size, the phenomenon of model overfitting will be more serious, so we chose the 30m grid for better prediction results.

***Thank you very much for your insightful and detailed comments.***